# Dorsomedial and ventromedial prefrontal cortex lesions differentially impact social influence and temporal discounting

Zhilin Su[1]*, Mona M. Garvert[2], Lei Zhang[1,3,4], Todd A. Vogel[1,3,4], Jo Cutler[1,3,4], Masud Husain[5,6,7], Sanjay G. Manohar[5,6,7], Patricia L. Lockwood[1,3,4,5,6]*

1 Centre for Human Brain Health, School of Psychology, University of Birmingham, Birmingham, United Kingdom, 2 Faculty of Human Sciences, Julius-Maximilians-University Würzburg, Würzburg, Germany, 3 Institute for Mental Health, School of Psychology, University of Birmingham, Birmingham, United Kingdom, 4 Centre for Developmental Sciences, School of Psychology, University of Birmingham, Birmingham, United Kingdom, 5 Department of Experimental Psychology, University of Oxford, Oxford, United Kingdom, 6 Wellcome Centre for Integrative Neuroimaging, University of Oxford, Oxford, United Kingdom, 7 Nuffield Department of Clinical Neurosciences, University of Oxford, Oxford, United Kingdom

* zhilinsu1312@gmail.com, z.su.1@pgr.bham.ac.uk (ZS); p.l.lockwood@bham.ac.uk (PLL)

## Abstract

The medial prefrontal cortex (mPFC) has long been associated with economic and social decision-making in neuroimaging studies. Several debates question whether different ventral mPFC (vmPFC) and dorsal mPFC (dmPFC) regions have specific functions or whether there is a gradient supporting social and nonsocial cognition. Here, we tested an unusually large sample of rare participants with focal damage to the mPFC ($N$ = 33), individuals with lesions elsewhere ($N$ = 17), and healthy controls ($N$ = 71) (total $N$ = 121). Participants completed a temporal discounting task to estimate their baseline discounting preferences before learning the preferences of two other people, one who was more temporally impulsive and one more patient. We used Bayesian computational models to estimate baseline discounting and susceptibility to social influence after learning others' economic preferences. mPFC damage increased susceptibility to impulsive social influence compared to healthy controls and increased overall susceptibility to social influence compared to those with lesions elsewhere. Importantly, voxel-based lesion-symptom mapping (VLSM) of computational parameters showed that this heightened susceptibility to social influence was attributed specifically to damage to the dmPFC (area 9; permutation-based threshold-free cluster enhancement (TFCE) $p$ < 0.025). In contrast, lesions in the vmPFC (areas 13 and 25) and ventral striatum were associated with a preference for seeking more immediate rewards (permutation-based TFCE $p$ < 0.05). We show that the dmPFC is causally implicated in susceptibility to social influence, with distinct ventral portions of mPFC involved in temporal discounting. These findings provide causal evidence for sub-regions of the mPFC underpinning fundamental social and cognitive processes.

**Data availability statement:** Data and code for modeling and analysis are openly available at the open science framework (OSF): https://osf.io/qzurp/. Unthresholded statistical maps generated in this study are available at https://identifiers.org/neurovault.collection:19609.

**Funding:** P.L.L was supported by a Medical Research Council Fellowship (MR/P014097/1 and MR/P014097/2; https://www.ukri.org/councils/mrc/), a Jacobs Foundation Research Fellowship (https://jacobsfoundation.org/), a Sir Henry Dale Fellowship funded by the Wellcome Trust and the Royal Society (223264/Z/21/Z; https://wellcome.org/, https://royalsociety.org/), a UKRI/EPSRC Frontier Science Guarantee (ERC Starting Grant Replacement Funding, EP/X020215/1; https://www.ukri.org/councils/epsrc/) and a Leverhulme Prize from the Leverhulme Trust (PLP-2021-196; https://www.leverhulme.ac.uk/). M.H. was supported by Wellcome Trust Principal Fellowship (098282/Z/12/Z, 206330/Z/17/Z; https://wellcome.org/) and NIHR Oxford Health Biomedical Research Centre funding (https://www.nihr.ac.uk/). S.G.M. was supported by a Clinician Scientist Fellowship (MR/P00878/X; https://www.ukri.org/councils/mrc/) and Leverhulme Research Grant (2018-310; https://www.leverhulme.ac.uk/). This research was also supported by the National Institute for Healthcare Research (NIHR; https://www.nihr.ac.uk/) Oxford Biomedical Research Centre (BRC; https://oxfordbrc.nihr.ac.uk/). L.Z. was partially supported by a Wellcome Data Science Ideathon Award (228268/Z/23/Z; https://wellcome.org/). Z.S. was supported by the Government Scholarship of Overseas Study funded by the Ministry of Education in Taiwan (https://english.moe.gov.tw/mp-1.html). The funders had no role in study design, data collection and analysis, decision to publish or preparation of the manuscript.

**Competing interests:** The authors have declared that no competing interests exist.

**Abbreviations:** AMI, Apathy-Motivation Index; ANCOVA, analysis of covariance; ANOVA, analysis of variance; BDI, Beck Depression Inventory; cTBS, continuous theta-burst stimulation; $D_{KL}$, Kullback–Leibler divergence; dmPFC, dorsomedial prefrontal cortex; FSL, FMRIB Software Library; HC, healthy controls; HMC, Hamilton Monte Carlo; KT, preference-temperature; KU, preference-uncertainty; LC, lesion controls; LMM, linear mixed-effects model; MCMC, Markov Chain Monte Carlo;

## Introduction

The medial prefrontal cortex (mPFC) has long been linked to processing social information and to economic decision-making [1–4]. Several studies have suggested that dorsal portions of mPFC (dorsomedial prefrontal cortex, dmPFC) are involved in processing social information [5–14], while ventral parts (ventromedial prefrontal cortex, vmPFC) are relatively more specialized in processing information pertinent to the self [7,15–22]. However, these conclusions have often been based on functional neuroimaging studies, which are correlational by nature, and the specificity of these different regions in social and economic processing is a topic of several ongoing debates [23].

Another perspective on the role of the mPFC in decision-making is that there is a spatial gradient along the ventral-dorsal axis purportedly distinguishing between self-referential (nonsocial) and other-regarding (social) processing. However, this division between self and others has also faced both theoretical and empirical challenges [23–25]. The vmPFC, including areas 11, 13, and 14, which is purported to be involved in processing self-relevant information (e.g., reflection about one's own personality traits [26]), has been shown to play a role in learning others' economic preferences [27], making choices for others based on their own preferences [23], integrating subjective values of self and others [28], and tracking the association between agents and objects for others [29]. On the other hand, the dmPFC (including area 9), presumed to be pivotal for social cognition, has been observed to engage in merging self- and other-related information [14,30] and representing one's own subjective values of choices during decision-making processes [31–33]. One interpretation of the results of these neuroimaging studies is that neither the vmPFC nor the dmPFC are specifically activated by social or nonsocial information. Causal evidence in large samples is strongly needed to reveal the necessity of the mPFC and its subdivisions in social cognition and economic decision-making.

Social and economic decision-making can be evaluated in parallel using paradigms such as the delegated inter-temporal choice task [23,27,34–36]. Humans and other animals differ significantly in their preference for immediate versus delayed rewards [37,38]. Some people are impulsive and have a strong preference for immediate rewards, even when they are smaller than those available in the future. In a temporal discounting task, participants are asked whether they prefer smaller sooner over larger later rewards [39]. By varying the values of these different rewards and fitting computational models we can precisely parametrize people's economic preferences for impulsivity versus patience. Strikingly, recent evidence suggests that such idiosyncratic preferences for future rewards can also be readily transmitted through social influence. When participants are tasked with making inter-temporal choices on behalf of someone else (i.e., delegated inter-temporal choices), they often adjust their own preferences to align with those of the other person [23,27,34–36]. This tendency to be influenced by others is a case of social influence or social contagion [5,12,40–42].

Existing work on the neural basis of social influence suggests regions of the mPFC may be crucial. A coordinate-based meta-analysis of functional neuroimaging studies suggested that activation of the mPFC (especially dorsal posterior parts) predicts people's conformity to a majority opinion [43]. Another neuroimaging study that evaluated the role of the mPFC in processing social information and economic decision-making linked activation of the dmPFC to conforming to a social norm, and activation of the vmPFC to social conformity and economic decision-making [10]. Finally, the process of shifting one's own preference to that of others could be driven by the plasticity of value representations in the mPFC. Indeed, a repetition suppression study showed a region in the mPFC where activity predicted susceptibility to social influence [27]. However, the causal necessity of the mPFC remains unknown.

mPFC, medial prefrontal cortex; TFCE, threshold-free cluster enhancement; TMT, Trail Making Test; VLSM, voxel-based lesion-symptom mapping; vmPFC, ventromedial prefrontal cortex.

Moreover, other studies instead point to the mPFC being involved in nonsocial decision-making. For example, activity of the mPFC [44] and its functional connectivity with other regions [45] have been shown to correlate with temporal discounting decisions. A handful of lesion studies have shown that damage to the mPFC had a null effect on temporal discounting [46] or led to an increase in temporal discounting [47–49]. Nevertheless, these lesion studies were conducted in fewer than 10 participants, and for neuroimaging studies, it is well-known that ventral portions of the mPFC are prone to considerable signal dropout due to their adjacency to bone and air sinuses, which might compromise the accuracy of functional localization within this area [50,51]. Taken together, these studies highlight the importance of using suitable causal approaches in large lesion samples to isolate if mPFC integrity is necessary for social influence and economic decision-making.

Here, we assessed the causal role of the mPFC in people's temporal discounting preferences and susceptibility to social influence, focusing on the nature of influence (i.e., being more impulsive or patient). We compared an unusually large group of rare participants with focal lesions to the mPFC ($N = 33$; Fig 1a) against two other control groups: participants with brain damage elsewhere (lesion controls, LCs; $N = 17$; Fig 1b) and age- and gender-matched participants without any brain damage (healthy controls, HC; $N = 71$). All participants first participated in a temporal discounting task designed to measure their baseline individual temporal discounting preferences. After completing this task, they were introduced to the preferences of two other people, being ostensible and unknown to the participants. The decisions of these two other people were in fact simulated based on a hyperbolic discounting model. One of these others was manipulated to have preferences that were more impulsive and the other as more patient, relative to the participants' estimated baseline preferences. Finally, participants completed the same temporal discounting task again (see Methods and Fig 2a) to examine whether learning the others' preferences resulted in social influence on their own discounting preference. To accurately estimate participants' temporal preferences and quantify their changes in preferences, we used a novel computational neurology approach fitting models to the data using hierarchical Bayesian modeling, and using the resulting parameters in lesion-symptom mapping.

We show that damage to the mPFC increases susceptibility to social influence. Crucially, those with mPFC lesions are more likely to be influenced by impulsive others compared to HCs, and more susceptible to social influence overall than LCs. Lesion-symptom mapping reveals that damage to the dmPFC (including area 9), and not to the vmPFC, is associated

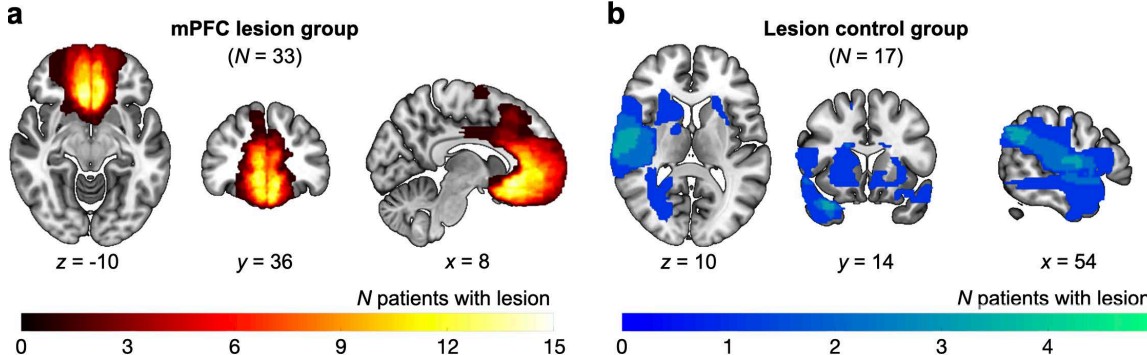

**Fig 1. Lesion locations for mPFC and lesion control groups.** (a) Participants in the mPFC lesion group ($N = 33$) had focal damage to the mPFC with the lesions extending into the lateral sections (area 13) of the bilateral mPFC and including medial surface subregions (areas 9, 14, 25, and 32). (b) Participants in the lesion control group ($N = 17$) also suffered damage mostly caused by subarachnoid hemorrhage but to areas outside the mPFC (see Methods). Note that the images here are shown in radiological convention.

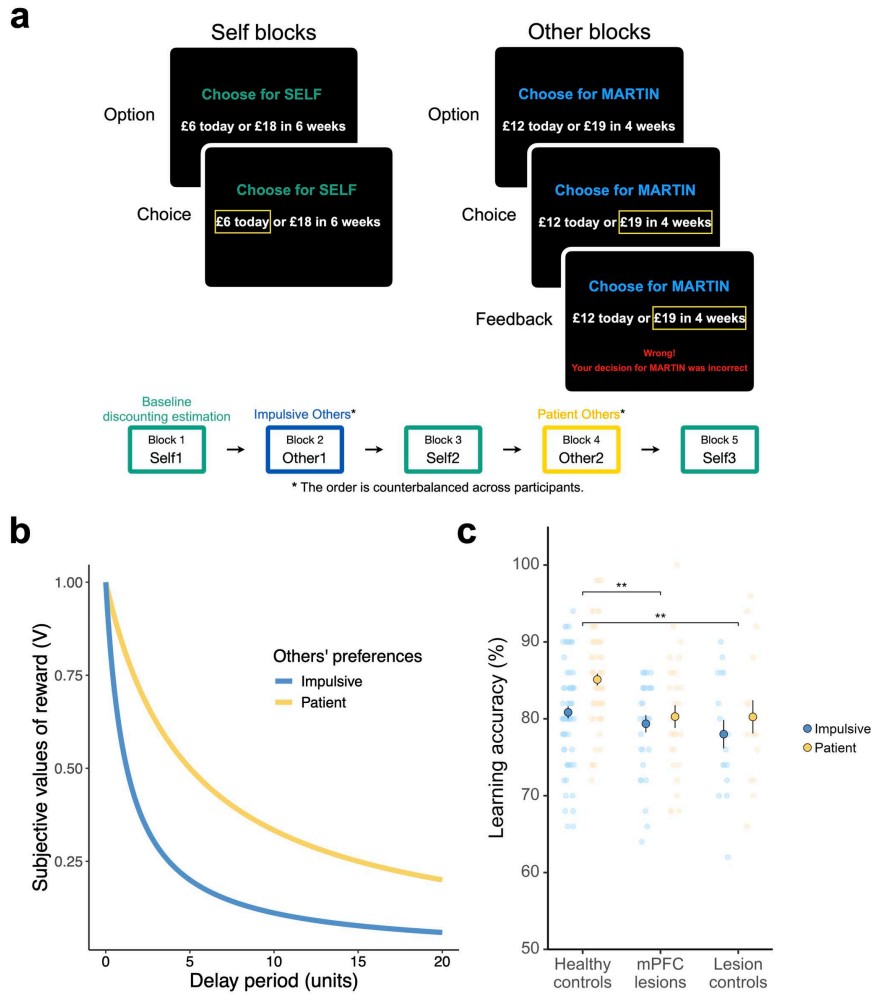

**Fig 2. The delegated inter-temporal choice task and learning performances. (a)** The trial structure in *Self* and *Other* blocks. During *Self* trials, participants were instructed to choose between two options: one offering an immediate smaller reward (smaller-and-sooner option, *SS*), and the other promising a larger reward after a variable delay period (larger-and-later option, *LL*). They were encouraged to express their genuine preferences by being informed that one of these choices would be randomly selected at the end of the study and serve as their bonus payment. During *Other* trials, participants were tasked to learn about the preferences of the other two people, with the information that these choices had been previously made by different participants. Participants were given feedback on their decisions, allowing them to grasp the intertemporal preferences of the other people. The experiment consisted of five blocks of 50 trials each (*Self1*, *Other1*, *Self2*, *Other2*, *Self3*), with a self-paced break after every 25 trials within each block, resulting in 250 trials overall. The order of the other people's preferences (*more impulsive* vs. *more patient*) was counterbalanced across participants. **(b)** Illustration of simulated hyperbolic discounters. The decisions of the other people were generated using a simulated hyperbolic discounting model (preference-temperature KT model, see Methods), where the discount rate *k* was adjusted to be either plus one (more impulsive) or minus one (more patient) from the participant's own baseline *k* in the first experimental block. **(c)** Participants with brain damage can accurately learn others' preferences. All three groups of participants (healthy controls, mPFC lesions, and lesion controls) were capable of learning in this task (right-tailed exact binomial tests against 50%, all $p$s < 0.001). Big circles with bordered lines represent the mean and error bars are the standard error of the mean, dots are raw data, and the asterisks represent the significant main effects of groups from the linear mixed-effects model and posthoc comparison. Note that the vertical axis starts from 50%, the chance level. **$p$ < 0.01. The underlying data and code used to generate this figure can be found at https://osf.io/qzurp/.

with an increase in susceptibility to impulsive social influence. Additionally, damage to both mPFC and damage elsewhere leads to greater baseline discounting compared to HCs. This heightened temporal discounting is associated with vmPFC (areas 13 and 25) and ventral striatum damage. Together, these findings reveal that the dmPFC is causally involved in social influence whereas the vmPFC is associated with temporal impulsivity.

## Results

To test the causal role of the mPFC in people's susceptibility to social influence and economic decision-making, we analyzed data from three groups: 33 participants with focal mPFC damage (mean age = 56.88; 17 females), 17 LC participants with brain damage not involving the mPFC (mean age = 56.24; 12 females), and 71 HC participants without any brain lesion (mean age = 60.73; 41 females). All participants first engaged in an inter-temporal choice task to assess their baseline individual discounting preferences. Following this, they were introduced to the preferences of two other players who they were informed had participated in the same temporal discounting task previously. They were instructed to learn these players' preferences through a trial-and-error process based on the feedback they received. In fact, these players were modeled to contrast with the participants' own tendencies (see Methods). One person was more impulsive, and one was more patient, relative to the participants' estimated baseline preferences (see Fig 2b). The decisions of the two other players were presented in a counterbalanced order across participants (see Fig 2a and Methods for more details). They also completed a series of neuropsychological tests, self-report measures of depression and apathy, and self-reported their perceived similarity to both impulsive and patient others at the end of the experiment. The three groups were closely matched, displaying no significant differences in terms of age, gender, visual attention, and executive function. Additionally, the two lesion groups showed no differences between each other in education, depression, or apathy (see Methods and S1 Table). Controlling for depression or apathy did not change any of our key results regarding group differences in temporal discounting or susceptibility to social influence (S1 Text and S2–S3 Tables).

Participants received feedback on their decisions, which allowed them to learn about the intertemporal preferences of the other people (see below *Simulation of the other people's choices*). The correct choices were characterized as those with greater estimated values from the hyperbolic model, based on a given discount rate. Due to the adaptive nature of the task, two HC participants and two mPFC participants had two others with '*more patient*' preferences. Data from these participants was therefore not available for analyses involving others with '*more impulsive*' preferences (i.e., learning accuracy and susceptibility to social influence). Similarly, eight HC participants, five mPFC participants, and one LC participant had two others with '*more impulsive*' preferences. Their data was not available for all analyses regarding others with '*more patient*' preferences (i.e., learning accuracy and susceptibility to social influence).

### Participants with brain damage can accurately learn others' preferences

To confirm participants were able to complete the task, our first analysis assessed their capacity to learn about the preferences of the other people who exhibited different discounting behaviors (Fig 2c). All three groups of participants (HC, mPFC, LC) demonstrated learning performances surpassing the chance level when learning about impulsive (HC mean [SE] = 81% [0.8%], mPFC = 79% [1.1%], LC mean = 78% [1.8%]; right-tailed exact binomial test against 50%, all proportions = 1.00, $ps < 0.001$) and patient others (HC = 85% [0.7%], mPFC = 80% [1.4%], LC = 80% [2.1%]; right-tailed exact binomial test against 50%, all proportions =

1.00, $ps < 0.001$). This suggests participants with brain damage, whether within the mPFC or elsewhere, were capable of learning others' preferences.

Next, we examined whether learning performances differed based on others' preferences among the three groups using a linear mixed-effects model (LMM; S2 Table). Overall, regardless of whether those preferences were more impulsive or patient, HCs demonstrated higher accuracy in learning others' preferences compared to the mPFC lesion group (main effect HC versus mPFC, $b$ [95% CI] = 3.22 [1.03,5.42], $p = 0.004$), while LCs performed similarly to the mPFC lesion group, with substantial Bayesian evidence of nonsignificant difference (main effect LC versus mPFC, $b$ [95% CI] = −0.66 [−3.71,2.38], $p = 0.67$, $BF_{01} = 3.32$). In addition, HCs were also more accurate in learning others' preferences than LCs (posthoc comparison HC versus LC estimate = 3.89, SE = 1.39, $t = 2.81$, $p = 0.006$). Therefore, participants with brain damage to the mPFC could learn others' preferences with high accuracy, although overall accuracy was lower than that of HCs and equivalent to LCs.

## mPFC lesions increase impulsivity but not uncertainty at baseline

After validating that all participants could successfully complete the task, we applied computational models of hyperbolic discounting [52,53], a widely used approach for indexing temporal discounting behavior. We utilized a previously validated Bayesian hyperbolic preference-uncertainty (KU) model to quantify participants' temporal impulsivity and choice uncertainty (Fig 3a, see Methods). The KU model proposes that participants' discounting preferences are best represented as a distribution, rather than a singular, fixed value [34]. The model was fitted through hierarchical Bayesian modeling [52,54] and verified using parameter recovery. The free parameters in the chosen model, $km$ (temporal impulsivity) and $ku$ (preference uncertainty), representing the mean and standard deviation of the participant's discounting distribution, exhibited excellent parameter recovery (all $r_s > 0.87$; S1 Fig). Additionally, the posterior predictive prediction successfully replicated the key patterns observed in our behavioral data (see Methods and S2 Fig). We therefore used this model to estimate participants' baseline discounting preference, and to determine whether these parameters varied between groups (Fig 3b).

Comparing the temporal impulsivity parameter (i.e., $km$) between groups revealed a main effect of group (one-way analysis of variance [ANOVA]: $F_{(2, 118)} = 6.36$, $p = 0.002$, $\eta^2$ [95% CI] = 0.10 [0.02,0.20]; S1 Text). We found that brain damage, whether within the mPFC or outside of it, resulted in increased temporal impulsivity compared to the HC group (posthoc comparison mPFC versus HC estimate = 1.30, SE = 0.40, $t = 3.23$, $p = 0.002$; LC versus HC estimate = 1.17, SE = 0.52, $t = 2.27$, $p = 0.03$). There was no significant difference in terms of temporal impulsivity between the two lesion groups (mPFC versus LC estimate = 0.13, SE = 0.57, $t = 0.22$, $p = 0.824$, $BF_{01} = 3.29$). Additionally, comparing the preference uncertainty parameter (i.e., $ku$) between groups also showed a main effect of group (one-way ANOVA: $F_{(2, 118)} = 9.55$, $p < 0.001$, $\eta^2$ [95% CI] = 0.14 [0.04,0.25]; S1 Text). While LCs demonstrated higher uncertainty in their own discounting preferences compared to HCs (LC versus HC estimate = 0.58, SE = 0.13, $t = 4.37$, $p < 0.001$), participants with mPFC lesions did not exhibit this behavioral pattern (mPFC versus HC estimate = 0.10, SE = 0.10, $t = 0.95$, $p = 0.343$, $BF_{01} = 2.81$). Even upon directly comparing the two lesion groups, LCs still showed greater preference uncertainty compared to those with mPFC lesions (LC versus mPFC estimate = 0.48, SE = 0.15, $t = 3.28$, $p = 0.001$). Notably, this increased preference uncertainty was not explained by total lesion size (correlation $ku$ versus lesion size within the LC group: $r_{s(15)} = −0.07$ [−0.54,0.42], $p = 0.779$, $BF_{01} = 3.24$). These findings suggest damage to the mPFC increases temporal impulsivity but not preference uncertainty.

## Damage to mPFC enhances susceptibility to impulsive social influence

After assessing participants' initial temporal preferences among groups, we proceeded to examine their susceptibility to social influence using signed Kullback–Leibler divergence ($D_{KL}$) (see Methods). $D_{KL}$ quantifies the difference between two probability distributions [35,55]. This metric evaluates the entire probability distribution, rather than solely focusing on summary statistics or point estimates derived from those distributions. We used $D_{KL}$ to formally quantify the shift of model parameters (i.e., *km* and *ku*) due to social influence (see Methods).

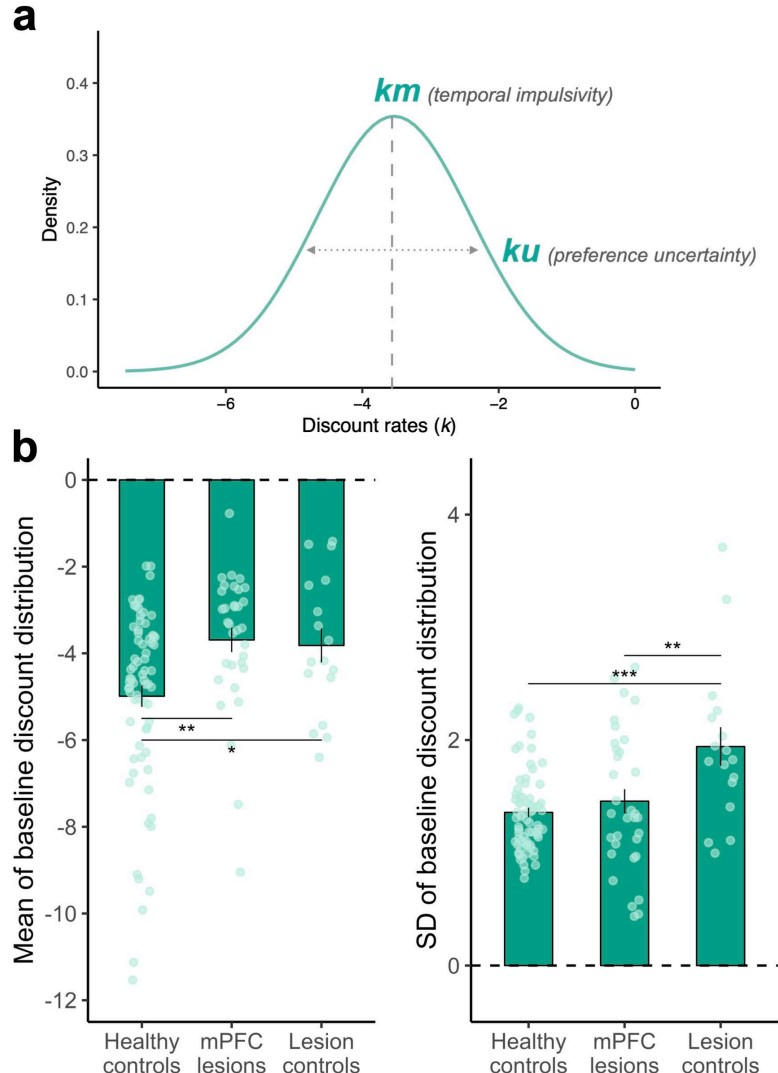

**Fig 3. mPFC lesions increase temporal impulsivity without affecting preference uncertainty. (a)** Illustration of the preference-uncertainty (KU) model. In the KU model, people's temporal discounting preferences are represented by a probability distribution. The mean (*km*) of this distribution indicates temporal impulsivity, while the standard deviation (*ku*) reflects the level of preference uncertainty. **(b)** Comparing temporal impulsivity *(km)* and preference uncertainty (*ku*) of participants derived from the preference-uncertainty (KU) model across groups revealed that mPFC lesions increased temporal impulsivity but not preference uncertainty compared to healthy controls. $N = 71$ for HC, $N = 33$ for mPFC, and $N = 17$ for LC. Bars show group means, error bars are standard errors of the mean, dots are raw data, and asterisks represent significant posthoc comparisons. *$p < 0.05$; **$p < 0.01$; ***$p < 0.001$. The underlying data and code used to generate this figure can be found at https://osf.io/qzurp/.

Throughout our analysis, we signed $D_{KL}$ to indicate the direction of shifting in the discounting distributions relative to the baseline. Positive signed $D_{KL}$ values signify a shift toward the discounting preferences of others (i.e., becoming more similar to others), whereas negative values indicate a divergence from them compared to baseline preferences.

We examined whether there were group differences in susceptibility to social influence when exposed to information about impulsive and patient others using an LMM (Fig 4 and S3 Table). Given differences in people's baseline impulsivity among the three groups, this LMM included participants' baseline *km* (continuous covariates, centered around the grand mean) and its interaction with fixed effects of group (*HC*, *mPFC*, and *LC*), other's preference (*patient* versus *impulsive*), and their interactions as fixed terms, along with a random subject-level intercept.

Strikingly, we found that participants with mPFC damage were more influenced by impulsive relative to patient others, compared to HCs (group × others interaction HC versus mPFC: *b* [95% CI] = 0.28 [0.03,0.54], *p* = 0.031). Posthoc tests uncovered that this interaction was primarily driven by the mPFC lesion group being more susceptible to impulsive social influence compared to HCs (HC versus mPFC estimate = −0.46, SE = 0.17, *t* = −2.70, *p* = 0.007).

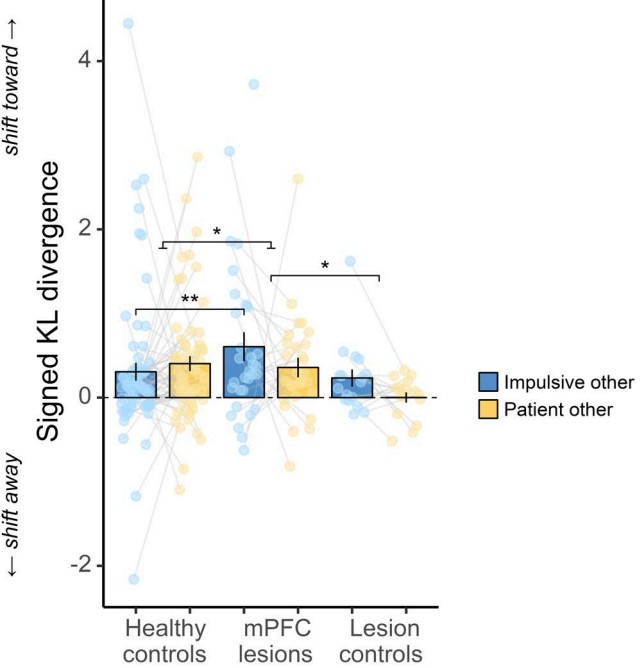

**Fig 4. Damage to mPFC increases susceptibility to impulsive social influence.** Compared to healthy controls, participants with mPFC lesions were more influenced by impulsive social influence (posthoc *p* = 0.007, follow-up of a significant LMM interaction). In contrast, the mPFC lesion group did not significantly differ from healthy controls in their susceptibility to patient social influence (posthoc *p* = 0.683, $BF_{01}$ = 4.01). Participants with mPFC lesions also showed heightened susceptibility to social influence overall, regardless of whether the influence was more impulsive or more patient, when compared to lesion controls (main effect mPFC vs. LC, *b* [95% CI] = 0.41 [0.05,0.77], *p* = 0.026). Sample sizes differ across conditions due to the adaptive nature of the task (*N* = 69 for HC impulsive, *N* = 63 for HC patient, *N* = 31 for mPFC impulsive, *N* = 28 for mPFC patient, *N* = 17 for LC impulsive, *N* = 16 for LC patient). Bars show group means, error bars are standard errors of the mean, and dots are raw data. Dots without connecting lines indicate participants with data unavailable for one of the two other players (see Methods). The asterisk between HC and mPFC represents the significant LMM interaction, while the asterisk between mPFC and LC indicates the significant LMM main effect. Asterisks between two impulsive bars signify a significant post-hoc comparison. **p* < 0.05; ***p* < 0.01. The underlying data and code used to generate this figure can be found at https://osf.io/qzurp/.

In contrast, there was no statistical difference between participants with mPFC lesions and HCs in their susceptibility to patient social influence (HC versus mPFC estimate = 0.08, SE = 0.19, $t$ = 0.41, $p$ = 0.683, $BF_{01}$ = 4.01). The mPFC lesion group was also overall more susceptible to social influence compared to LCs (main effect LC versus mPFC, $b$ [95% CI] = −0.41 [−0.77 −0.05], $p$ = 0.026). Additionally, we re-ran the analysis to confirm that results remained the same accounting for the order of others' preferences (see S4 Table), and no significant correlation was found between impulsive and patient signed KL divergence in any group ($p$s > 0.49, S5 Table), suggesting that the order effect could not explain the group differences observed here. Furthermore, an exploratory control analysis that accounted for baseline preference uncertainty did not change the interaction results reported above (S6 Table), suggesting that the group differences in susceptibility to social influence were not attributed to individual differences in preference uncertainty. Importantly, although participants with mPFC lesions were relatively more susceptible to impulsive social influence, they did not report feeling more similar to impulsive others (main effect patient others versus impulsive others on perceived similarity within mPFC lesions: $b$ [95% CI] = 0.35 [−0.08,0.78], $p$ = 0.107, $BF_{01}$ = 1.28), with anecdotal Bayesian evidence suggesting no difference. Their susceptibility to social influence was also not correlated with their learning performances ($p$s > 0.83, see S7 Table) or with their perceived similarity to others ($p$s > 0.12, S8 Table), suggesting these group differences were not driven by possible individual differences in learning ability or perceived similarity to others. Taken together, these results demonstrate that brain damage specifically to the mPFC enhanced people's susceptibility to social influence, with impulsive social influence particularly affected.

## Damage specifically to dmPFC is associated with heightened susceptibility to impulsive social influence

Next, we used voxel-based lesion-symptom mapping (VLSM) to examine whether subregions within mPFC were linked to group differences in susceptibility to social influence. The VLSM analysis pinpoints voxels where participants with damage at that voxel, compared to participants with damage elsewhere, show differences in susceptibility to impulsive relative to patient social influence (i.e., signed impulsive $D_{KL}$ minus signed patient $D_{KL}$; $N$ = 26 where both patient and impulsive others were present, see Methods). VLSM assesses whether the lesion in each voxel predicts an individual's behavior by generating a map of the $t$-statistics [56]. We included voxels where damage was present in at least five participants [57]. We used the FMRIB Software Library (FSL) [58] to conduct permutation-based VLSM with threshold-free cluster enhancement (TFCE) [59,60]. The combination of permutation testing with TFCE allowed us to achieve an optimal balance between sensitivity to true effects and reducing the risk of identifying small, potentially spurious effects [56,59]. Significance was reported at permutation-based TFCE $p$ < 0.025 (permutation-based TFCE $p$ < 0.05 Bonferroni-corrected across two behavioral regressors). As a control analysis, we first confirmed that there was no significant association between the overall degree of damage (i.e., total lesion size) and susceptibility to social influence (see Methods).

The VLSM analysis revealed only one region, in the dmPFC incorporating parts of area 9 (Fig 5, peak MNI coordinate [±2, 40, 20], cluster size $k$ = 282), that correlated with the behavioral difference in susceptibility to social influence. To further examine this correlation between this dmPFC area and increased susceptibility to impulsive social influence, we repeated our analysis incorporating LCs with damage outside of the mPFC ($N$ = 42 in total). This analysis confirmed the involvement of an overlapping region within the dmPFC (area 9; S3 Fig, peak MNI coordinate [±2, 40, 20], cluster size $k$ = 1) identified in our prior analysis.

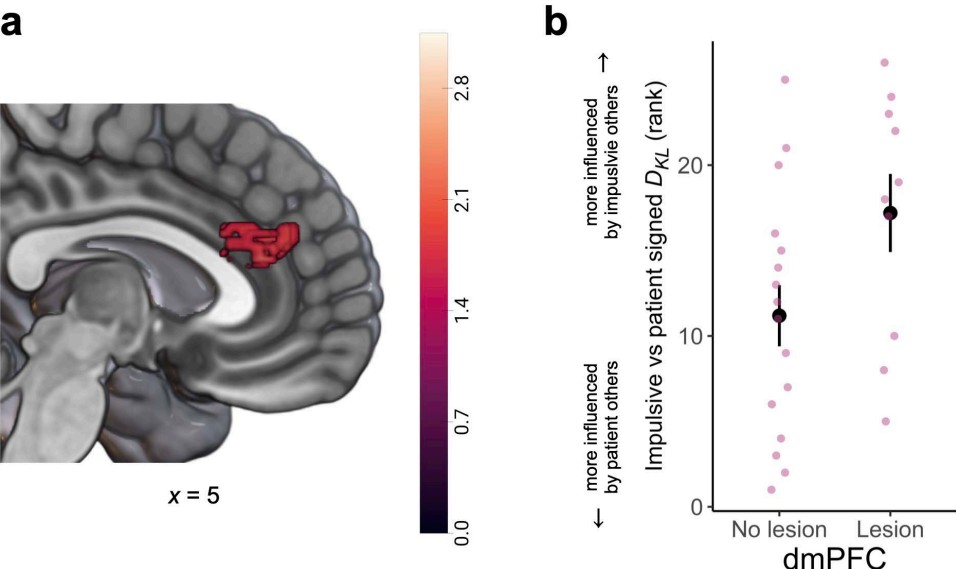

**Fig 5. Damage to dmPFC (area 9) enhances susceptibility to impulsive social influence. (a)** Permutation-based, whole-brain, nonparametric voxel-based lesion-symptom mapping (VLSM) showed that damage to dorsomedial prefrontal cortex (dmPFC, area 9) was associated with heightened susceptibility to impulsive relative to patient social influence (permutation-based threshold-free cluster enhancement (TFCE) $p < 0.025$). **(b)** Plotting the ranked contrasts between susceptibilities to impulsive and patient social influence, separately for participants with damage or no damage in the areas identified by the VLSM analysis. $N = 26$ for this analysis where data from patient and impulsive was present. The underlying data and code used to generate this figure can be found at https://osf.io/qzurp/. Note: panel (b) is for illustrative purposes only and displays the ranked difference in signed KL divergence contrasts between participants with vs. without lesions, in the ROI defined by a wholebrain contrast.

These results highlight that damage to an area within the dmPFC, rather than ventral portions, made people more susceptible to influence by impulsive versus patient others.

## Damage to vmPFC and ventral striatum is associated with increased temporal impulsivity

Finally, we used another VLSM to test whether there were any mPFC subregions where damage underpinned the behavioral increase in temporal impulsivity, that is how much participants discounted the reward value over time (i.e., *km* parameters; $N = 33$ for mPFC lesion participants) (see Methods). Again, there was no significant association found between the overall degree of damage (i.e., total lesion size) and temporal impulsivity (see Methods). We found no significant correlation between mPFC damage and heightened baseline temporal impulsivity at our threshold criteria (permutation-based TFCE $p < 0.025$). Subsequently, we adopted an exploratory approach, examining whether any regions were significantly associated at uncorrected levels after permutation testing ($p < 0.05$). This analysis revealed that lesions in two distinct clusters, one encompassing the ventral portions of the mPFC corresponding to area 13 (Fig 6, peak MNI coordinate [±16, 14, −18], cluster size $k = 6$) as well as area 25 (peak MNI coordinate [±6, 18, −8], cluster size $k = 2$), and another in the most ventral parts of the striatum putatively corresponding to the nucleus accumbens (peak MNI coordinate [±12, 14, −12], cluster size $k = 2$). In these areas, damage was associated with increased temporal impulsivity, as evidenced by increased *km* parameters.

To provide further evidence for the robustness of this exploratory analysis, we repeated our analysis including LCs with damage outside of the mPFC ($N = 50$ in total). Here, we again

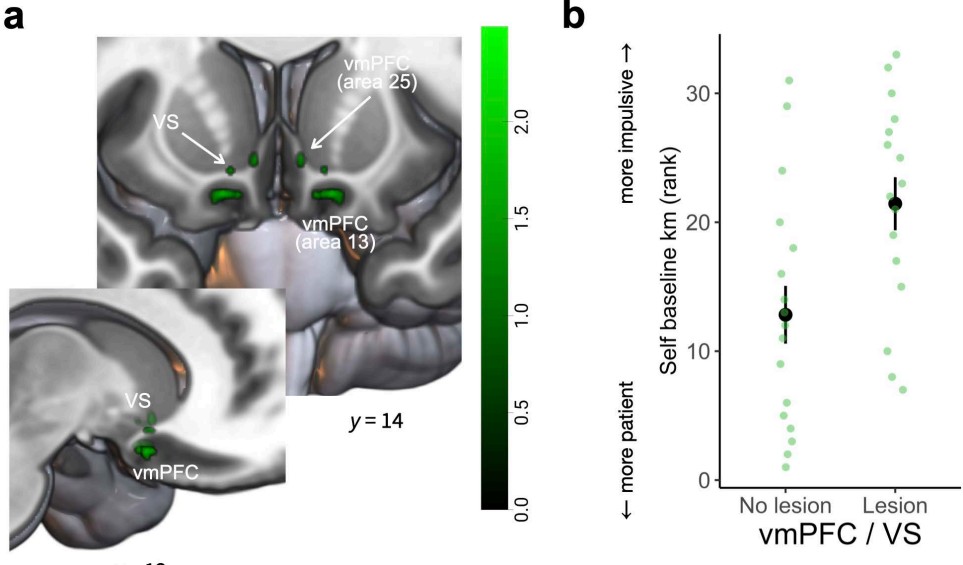

**Fig 6. Damage to vmPFC and ventral striatum increases temporal impulsivity. (a)** Permutation-based, whole-brain, nonparametric voxel-based lesion-symptom mapping (VLSM) showed that the areas 13 and 25 in the vmPFC as well as ventral striatum where damage was correlated with increased temporal impulsivity (permutation-based threshold-free cluster enhancement (TFCE) $p < 0.05$). **(b)** Plotting the ranked self baseline discounting preferences, separately for participants with damage or no damage in the areas identified by the VLSM analysis ($N = 33$). The underlying data and code used to generate this figure can be found at https://osf.io/qzurp/. Note: panel (b) is for illustrative purposes only and displays the ranked difference in self-baseline discounting preferences between participants with vs. without lesions, in the ROI defined by a wholebrain contrast.

found a portion in the vmPFC area 25 (S4 Fig, peak MNI coordinate [±6, 12, −10], cluster size $k = 3$) and ventral striatum (peak MNI coordinate [±14, 22, −4], cluster size $k = 3$) where damage was correlated with enhanced baseline temporal impulsivity.

## Discussion

Several lines of evidence implicate the mPFC as crucial for processing social information and for economic decision-making [1–4]. However, theoretical and empirical accounts of mPFC function have been mixed, with studies claiming a role in economic or social processing, or both, and precise contributions of distinct mPFC subregions often overlooked. Here, by integrating an economic decision-making task measuring susceptibility to social influence in parallel with temporal discounting and leveraging Bayesian computational models, we demonstrate the mPFC is causally involved in social influence. Moreover, heightened susceptibility to impulsive social influence is attributed to specific damage to the dmPFC. We also observed that mPFC damage was associated with increased baseline temporal discounting compared to HCs, with this heightened temporal impulsivity linked to damage in vmPFC and ventral striatum in exploratory analyses. Together, these results demonstrate the fundamental role of the dmPFC in social influence.

Previous neuroimaging studies have suggested that the dmPFC processes social conformity by detecting misalignment between one's own and other's opinions [1,12,43,61,62], with its activity associated with the extent of subsequent conformity under social influence [63–68]. A functional neuroimaging study on the social contagion of risk preferences also found that the dmPFC, along with the dorsolateral prefrontal cortex and inferior parietal lobule, was

involved in belief updating when participants learnt about others' risk preferences [69]. In addition, in both humans [70,71] and macaque monkeys [72], the dmPFC has been shown to track the reliability of social information and to moderate the integration of self and social information based on their respective levels of certainty. This belief updating mechanism holds significance in understanding social conformity. However, following this line of research, one might expect that damage to the dmPFC would lead to decreased susceptibility to social influence, rather than an increase [73,74]. Instead, we found that damage here increased susceptibility to social influence.

One putative function of the dmPFC is in maintaining self-other distinction [75], differentiating signals attributed to oneself from simulated signals attributed to another person [76,77]. Achieving successful self-other distinction is essential for effective social interaction, including optimal display of social conformity. Recent studies have suggested that the dmPFC facilitates distinguishing the abilities of others from one's own [30] and that applying continuous theta-burst stimulation (cTBS) over the dmPFC disrupts this self-other distinction [14]. Moreover, another study found that disrupting the dmPFC activity through transcranial ultrasound stimulation led macaque monkeys to exhibit suboptimal reliance on unreliable social information over nonsocial (self) information [72]. Similarly, a recent study found that downregulating the dmPFC activity using cTBS impaired learning performance during observational action-based learning by disrupting the predictability of the demonstrator's actions [78]. Therefore, one possibility is that damage to dmPFC could blur the self-other distinction and hinder the effective use of social information, prompting people to excessively depend on others for information, thereby increasing susceptibility to social influence. This process may drive the asymmetry we observed in susceptibility to impulsive versus patient social influence in those with dmPFC damage. Although they did not report feeling more similar to impulsive others, their similarity to impulsive others could drive an implicit process where they were particularly susceptible to being influenced by others who also displayed impulsive choices.

Another perspective on the mechanisms behind social conformity is related to reinforcement learning [66,68,79]. In the reinforcement learning framework, learning is driven by prediction errors, the discrepancy between expected and actual outcomes [80]. When people's own preferences differ from those of others, such social expectancy prediction errors are encoded in the dmPFC [27,63,66–68,81]. People use this error signal to reduce the difference between self and others by either learning from or conforming to others [63,66–68]. In situations where people are unable to fully know the preferences or intentions of others, but still consider others' choices to be informative, they must infer other's mental states to optimize their own actions. In such scenarios, they need to evaluate the reliability of others' choices, emulate others' intentions, and integrate the inferred social information with their own, all of which entail the involvement of the dmPFC [70,82]. Therefore, given the central role of the dmPFC in reinforcement learning within social contexts, damage to the dmPFC may result in atypical social prediction errors which heighten social conformity. Future studies could probe these alternative explanations further. Notably, learning accuracy of others' preference was intentionally high in the current paradigm to ensure all groups were able to learn others' preferences so they could be influenced by them. Future paradigms could explicitly measure the effects of dmPFC damage on social learning in paradigms where learning accuracy is more variable such as while assessing mentalizing [83], vicarious learning [13,71,84], or other social behaviors. In nonsocial decision-making, theories of the mPFC suggest that it may contextualize learning by providing a 'task space' or map that allows learning to be constrained to certain 'states' [85]—for example, the mPFC may prevent learning when generalizing to irrelevant contexts. Perhaps maintaining self-other distinction during learning could be regarded as a specific case of this.

It is somewhat surprising that people adjust their own preferences to align with others, even when such alignment could reduce their bonus payment. In our experimental design, participants were incentivized to prioritize their own outcomes and were explicitly informed that their decisions for others had no consequences for anyone involved, which highlights the robustness of the observed effects. It suggests that these preference shifts are not merely a byproduct of external factors, but instead reflect deeper cognitive or motivational processes [27,34].

The question of whether the brain has specialized regions and circuits for social behavior is central to social neuroscience [8,86–88]. Previous work has identified how social specificity may be realized at different levels of explanation [24]. Our task had many features to enhance its ability to capture social processes, including two different social others with different preferences, informing participants the choices they observed were from real others, and carefully probing for any disbelief in the social manipulation. Furthermore, existing studies including control conditions that match the same stimuli and actions, but do not require social simulation, failed to replicate changes in participants' discounting preferences [27]. This suggests that simulation of other agents' mental states—a central aspect of social interaction [88,89] — is essential for the observed changes in people's preferences, which highlights the importance of social component of the influence effect. To fully address whether shifts in people's own preferences occur in the absence of social influence, future studies could consider including a nonsocial control targeting different levels of explanation for social specificity. This additional control condition could reveal the cognitive boundaries and specific neural systems that underpin social influence and whether they are common or distinct from nonsocial processes.

Our findings show that damage to the mPFC results in heightened preferences for immediate reward options over delayed ones, aligning with prior findings suggesting the significant involvement of the mPFC in temporal discounting [44,45,47–49,90–95]. In addition to showing these robust effects at the group level, we exploratively localized heightened impulsivity to the vmPFC, putatively in areas 13 and 25. Prior studies have suggested that the vmPFC plays a crucial role in inter-temporal decision-making, with damage to the vmPFC (in smaller samples) typically resulting in increased temporal discounting [47–49,95]. A recent study also showed that individual differences in temporal discounting preferences could be predicted by specific patterns of brain activity involving the vmPFC [96]. One possible explanation for people's preference for immediate rewards over future ones is the less tangible and more abstract nature of future rewards [97,98]. It has been reported that vividly imagining prospective events (i.e., episodic future thinking) reduces temporal discounting [99,100], supporting the assertion that future rewards are less favored due to their perceived intangibility. The integrity of the vmPFC may be crucial in episodic future thinking [48,101–104].

We also found that damage to the ventral striatum, previously linked to processing value and reward [15], was associated with steeper temporal discounting. These findings are important as they provide initial causal evidence for the role of striatum in inter-temporal decision-making where its function is highly debated. While several human neuroimaging studies have linked the ventral striatum to encoding the subjective value of delayed rewards [17,105,106], reflecting the difference in subjective value between delayed and immediate rewards [107], as well as tracking the objective magnitude of delayed rewards [108], other evidence suggests that the ventral striatum may exhibit increased activation in response to immediate rewards compared to delayed ones [109,110]. Moreover, some studies have demonstrated that ventral striatum activity is positively associated with temporal impulsivity [44,111], whereas others have found that its activity tracks participants' choices for delayed rewards [107]. Due to the anatomical location of the ventral striatum, there have been limited lesion studies or noninvasive stimulation studies in humans. Intriguingly, research in rodents

has revealed that damage to the ventral striatum core results in a reduced probability of selecting delayed rewards [112,113], which fits with our finding here. Future studies could use new brain stimulation techniques, such as focused ultrasound, to dynamically module ventral striatum response during temporal discounting.

In addition to these novel findings, our study also has limitations. While we were able to recruit a relatively large sample over several years, there were fewer participants with damage covering ventral striatum and only exploratory evidence for a role of this area. Further studies in larger samples are needed to confirm the precise role of the ventral striatum in temporal discounting. Second, we measured a specific type of social influence in terms of economic preferences. It would be important for future work to map the wider types of social influence that are associated with dmPFC function. For example, the dmPFC and adjacent perigenual cingulate cortex have been linked to tracking confidence in several neuroimaging studies [114,115]. While we did not observe any group differences between those with mPFC lesions and HCs in processing uncertainty at baseline, it would be interesting to evaluate the role of confidence in being influenced by other people. In contrast, HCs differed from both lesion groups in their baseline temporal discounting preferences. However, we controlled for baseline discounting preferences in our statistical models, and the two others that participants learnt about were modeled to be more impulsive or patient relative to participants' own baseline. The two lesion groups also did not differ, despite having brain damage in distinct areas. This ensured that differences in initial temporal discounting, before social influence, were accounted for. Additionally, while we have used advanced lesion-symptom mapping with a relatively large cohort of patients to establish a causal link between the mPFC and susceptibility to social influence, there could be alternative explanations for some associations. For example, there could be shared causes that make brain lesions and impulsivity more likely to co-occur. However, the LC group was designed to control for effects that simply correlate with having brain lesions. Moreover, our choices of patients who predominantly had aneurysmal hemorrhages, which are stochastic events with relatively weak causal associations with impulsivity, also reduced the chance of a confounding variable influencing our findings. Future research would ideally take a multi-center longitudinal approach to be able to provide even stronger causal evidence.

In conclusion, we show that participants with damage to the mPFC are more prone to social influence. This increased susceptibility to social influence was linked to specific damage to the dmPFC when such influence was impulsive. Furthermore, lesions to the mPFC were associated with elevated baseline temporal discounting compared to HCs. This heightened temporal impulsivity was linked to lesions in the vmPFC and ventral striatum in exploratory analyses. Taken together, these results reveal that the mPFC plays a causal role in social influence with damage specifically to the dmPFC crucial for being influenced by others.

## Materials and methods

### Participants

Three groups of participants were recruited: the lesion group with focal damage to the mPFC, the LC group with lesions outside of the mPFC, and the age- and gender-matched HC group. The lesion participants were selected from a database of 453 individuals with neurological conditions, while the HCs were recruited from university databases and the community. The mPFC lesion group consisted of 33 patients with mPFC damage (age range = 37–76, mean = 56.88; 17 females). The LC group consisted of 17 participants with lesions in areas outside the mPFC (age range = 28–74, mean = 56.24; 12 females). The HC group consisted of 71

participants without any brain damage (age range = 24–76, mean = 60.73; 41 females), leading to a total sample of $N$ = 121 for behavioral analyses. Classification of lesion location was performed from MR imaging or CT scans by a clinical neurologist (SGM). All participants gave their written consent to participate in the study, which has been ethically approved by the Medical Sciences Interdivisional Research Ethics Committee at the University of Oxford (Approval number: 18/LO/2152). The study was conducted according to the principles expressed in the Declaration of Helsinki.

The majority of patients had suffered subarachnoid hemorrhages from rupture of an aneurysm (anterior communicating artery aneurysm in mPFC patients). Four had frontal meningiomas resected, and one had an ischemic stroke. The participants were carefully screened and selected to ensure there were no discrepancies in terms of gender ($\chi^2_{(2)}$ = 1.67, $p$ = 0.433) or age ($p$s > 0.20). In the mPFC group, 13 were on antihypertensives, two were taking amitriptyline, one was on pregabalin, and one was taking levetiracetam, with no other neurological or psychiatric medication. In the LC group, four were on antihypertensives, two were on citalopram, and one was on paroxetine, one was on pregabalin, one on pregabalin, and one was on lamotrigine plus levetiracetam. The mPFC lesion group also did not significantly differ from other controls in performance on a neuropsychological test assessing visual attention and executive function (Trail Making Test (TMT) [116]; Part A $p$s > 0.32, Part B $p$s > 0.19). However, they reported slightly higher levels of apathy (Apathy-Motivation Index (AMI) [117]; $p$ = 0.014) and depression (Beck Depression Inventory (BDI) [118]; $p$ = 0.033) compared to the HC group. There was no significant difference between these measures when comparing the mPFC group to the LC group ($p$s > 0.15).

One participant from the HC and mPFC groups had incomplete data on the self-report questionnaire measures, leading to their exclusion from the relevant analyses. In the final sample, as a result of the task's adaptive nature, two HC participants and two mPFC participants had two others with '*more patient*' preferences. Data from these participants were unavailable for analyses regarding others with '*more impulsive*' preferences (i.e., learning accuracy and susceptibility to social influence). Likewise, eight HC participants, five mPFC participants, and one LC participant had two others with '*more impulsive*' preferences. Their data was unavailable for all analyses related to others with '*more patient*' preferences (i.e., learning accuracy and susceptibility to social influence).

## Lesion identification

Of the 50 patients, all except two had MR imaging (1 mm isotropic T1 FSPGR MRI with 6 mm axial T2 PROPELLER sequence). Two cases had only a CT scan as they had metal surgical clips and an implantable defibrillator. Before conducting behavioral testing, a clinical neurologist (SGM) manually outlined each participant's lesion on their brain scan, utilizing FSL [58] (http://fsl.fmrib.ox.ac.uk/fsl) to map it onto the MNI152 template. Each lesion map was processed using a Gaussian kernel with a 5 mm full-width at half-maximum convolution. The average volume of the lesions was 2.68 cm³ (SD 2.63), and volume varied between 0.02 and 9.74 cm³. There was no statistically significant difference in lesion volumes between two lesion groups (mPFC mean [SD] = 2.28 [2.38]; LC mean [SD] = 3.47 [2.98]; $W$ = 210, $Z$ = −1.43, $r_{(48)}$ = 0.20 [0.01,0.48], $p$ = 0.153, $BF_{01}$ = 1.31). There was no significant correlation between the overall lesion volume and any of the variables included in the VLSM analysis (self baseline discount rates, contrasts between susceptibilities to impulsive and patient influences), either across all participants ($p$s > 0.13) or within the mPFC group ($p$s > 0.35). To illustrate the extent of the lesions, an overlap map was created by counting the number of participants with lesions exceeding 10% degree of lesions within each voxel (Fig 1a and 1b).

## Procedure

Participants took part in a one-time on-site test that began with a clinical assessment with a neurologist (SGM). Following this, participants completed the delegated inter-temporal choice task [35], in addition to three separate experimental tasks (being reported elsewhere) and a series of other questionnaires. Participants received compensation of £10 per hour and were informed they would earn an extra bonus determined by a trial randomly selected from the task: the bonus would be awarded following a designated delay period, unless immediately. Actually, participants received a bonus that varied between £1 and £10 chosen randomly on the day they were tested and were notified that a trial had been selected.

**Delegated inter-temporal choice task.** Participants engaged in a delegated inter-temporal choice task where they learnt about the preferences of impulsive and patient others after making their own temporal discounting choices (Fig 2a). During the task, participants were asked to choose between two options: one was a smaller amount of money delivered immediately (*today*), while the other was a larger amount of money delivered after a variable delay period. The amount of reward ranged from £1 to £20, and the delay period varied between 1 and 90 days (this was subject to dynamic adjustments in the *Self* blocks). Both the immediate and delayed options were displayed simultaneously, with their positions on the screen being randomized across trials. The whole experiment consisted of five blocks of 50 trials (*Self1*, *Other1*, *Self2*, *Other2*, *Self3*), with a self-paced break halfway through each block, resulting in 250 trials in total. Participants were told that the decisions they would learn about during the task were those made by prior participants of the study. However, in reality, these decisions were generated by a simulation algorithm (see Methods). None of the participants reported disbelief regarding the authenticity of these decisions being from actual people during or after the task to the experimenter. We further probed whether they had any disbelief in a post-study survey by asking if they had any questions or concerns about the task they completed. Both checks further demonstrated the validity of our task.

During the trials within the *Self* blocks (i.e., the first, third, and fifth blocks), participants were instructed to choose the option that genuinely reflected their own preferences, as they believed that one of these chosen options would be actualized as their bonus payment. During the trials within the *Other* blocks (i.e., the second and fourth blocks), participants were instructed to learn about the decisions made by two others, under the belief that these choices reflected the decisions of previous participants. The behaviors of these two people were simulated based on the participants' own decisions from the *Self1* block. Participants received feedback on their decisions, which allowed them to learn about the intertemporal preferences of the other people (see below *Simulation of the other people's choices*). The correct choices were characterized as those with greater estimated values from the hyperbolic model, based on a given discount rate. Two names, either gender-matched or randomly selected for participants who did not indicate their gender, were chosen to present the other two people. The participants were made aware that their selections on behalf of others were not relayed to those people and had no consequences for either themselves or the other people. The task was displayed using MATLAB 2012a (The MathWorks) and the Cogent 2000 v125 graphic toolbox, a software developed by the University College London, which was formerly accessible at www.vislab.ucl.ac.uk/Cogent/.

**Apathy Motivation Index.** The AMI [117], a scale consisting of 18 items, was used to assess participants' apathetic traits. This scale measures three dimensions of individual differences in apathy-motivation: behavioral activation, social motivation, and emotional sensitivity. Participants rated their agreement with each item on a 5-point Likert scale, ranging from 0 to 4. Each item's score is reversed, meaning that higher scores indicate increased levels of apathy.

**Beck Depression Inventory.** Symptoms of depression were assessed through the 21-item BDI [118]. Each item was rated by participants on a 4-point Likert scale from 0 to 3, with higher cumulative scores signifying increased severity of depressive symptoms.

**Trail Making Test.** The TMT [116], which includes two parts, is designed to be completed as swiftly and accurately as possible. In TMT-A, participants are tasked with sequentially drawing lines to connect 25 numbers scattered randomly on a paper in ascending order (i.e., 1–2–3–4, etc.), serving as a test of visual attention. TMT-B requires participants to alternate between numbers (1–13) and letters (A–L) in their connections (i.e., 1–A–2–B–3–C, etc.), which is considered a measure of executive function. The time taken to finish each part of the test is recorded as the score.

**Delegated inter-temporal choice task-specific questionnaires.** Participants were presented questions to assess their perceived similarity to others in the task. They provided their ratings using a sliding scale ranging from 0 (*not at all*) to 10 (*very similar*). All self-report measures were collected via the Qualtrics platform (https://www.qualtrics.com/).

## Statistical analysis

We used R [119] (v4.2.1) along with RStudio [120] (v2023.06.2+561) to analyze the data. Behavioral data and fitted model parameters (see below) were analyzed using LMMs ('*lmer*' function from the {*lme4*} package [121] v1.1-33) or linear regression ('*lm*' function from the {*stats*} package [119] v4.2.1).

LMMs were used to predict participants' learning accuracy, signed KL divergence, and self-report perceived similarity. These models incorporated fixed effects for group (*HCs*, *mPFC lesions*, and *LCs*), other's preference (*patient* versus *impulsive*), and their interaction, as well as a random intercept at the subject level. Considering the differences in temporal impulsivity at the baseline among the three groups, the LMM for signed KL divergence also included participants' baseline temporal impulsivity ($km$; continuous covariates, centered around the grand mean) and its interaction with groups and other's preferences (including the three-way interaction) as fixed terms. Additionally, control analyses of accuracy and signed KL divergence separately included the BDI and AMI scores as a fixed term, without interacting with the other terms (see below). A further control analysis was performed to examine the effect of the order of others' preferences on the signed KL divergence. An exploratory control analysis was conducted to account for individual differences in baseline preference uncertainty ($ku$; continuous covariates, centered around the grand mean). Simple linear regressions were used to compare the group differences in their age, education years, BDI scores, AMI scores, and TMT scores. One-way analyses of variance were used to compare the temporal impulsivity ($km$) and preference uncertainty ($ku$) parameters across groups. As control analyses, analyses of covariance (ANCOVA) that separately included BDI scores and AMI scores were conducted to control for the effects of depression and apathy levels.

The LMMs were set up as follows (note that each participant contributed a single parameter data point and therefore these models could not contain random slopes):

LMM1a: Accuracy ~ Group * Preference + (1|ID)
LMM1b: Accuracy ~ Group * Preference + BDI + (1|ID)
LMM1c: Accuracy ~ Group * Preference + AMI + (1|ID)
LMM2: Similarity ~ Group * Preference + (1|ID)
LMM3a: Signed KL divergence ~ Group * Preference * Self baseline impulsivity + (1|ID)
LMM3b: Signed KL divergence ~ Group * Preference * Self baseline impulsivity + BDI + (1|ID)
LMM3c: Signed KL divergence ~ Group * Preference * Self baseline impulsivity + AMI + (1|ID)
LMM3d: Signed KL divergence ~ Group * Preference * Self baseline impulsivity + order + (1|ID)

LMM3e: Signed KL divergence ~ Group * Preference * Self baseline impulsivity + Self baseline preference uncertainty + (1|ID)

Simple group comparisons were conducted using either independent parametric (*t-test*) or nonparametric (Wilcoxon two-sided signed rank test) methods. To assess nonsignificant results, Bayes factors ($BF_{01}$) were calculated using either paired and independent Bayesian *t*-tests ('*ttestBF*' function from the {*BayesFactor*} package [122] v0.9.12-4.4) or through linear models ('*lmBF*' function from the same package) with the default prior. $BF_{01}$ measures how much more likely that the data is under the null hypothesis of no difference, as opposed to the alternative hypothesis of a difference. The interpretation and reporting of Bayes factors followed the terminology recommended by Jeffreys [123]. All figures of statistical analysis were generated using the {*ggplot2*} package [124] (v3.4.2).

## Computational modeling

Participants' decisions in each experimental block were separately used to estimate their discount rates using a standard hyperbolic discounting model [53]:

$$V_{LL} = \frac{M_{LL}}{1 + KD} \tag{1}$$

where $V_{LL}$ represents the subjective value of a larger-and-later option, $M_{LL}$ denotes the objective magnitude of that reward, $D$ is the delay before receiving the reward, and $K$ is the hyperbolic discount rate specific to each participant, which quantifies the devaluation of larger-and-later options by time. The subjective value ($V_{SS}$) of a smaller-and-sooner option is always equivalent to its objective magnitude ($M_{SS}$) because the delay period for this reward is zero. Previous studies indicate that the parameter, $k = \log_{10}(K)$, usually follows a nearly normal distribution in the population [27,34]. Therefore, all the analyses presented are based on $k$, which is the log-transformed measure of $K$. As $k \to -\infty$, people generally do not discount delayed options, evaluating an offer purely on its objective magnitude. When $k \to 0$, people grow more sensitive to delay periods and tend to discount delayed options more steeply.

**Preference-temperature (KT) model.** In the course of the experiment, the preference-temperature (KT) model was applied to approximate participants' behaviors in the *Self1* block and to simulate the choices made by the other people. The KT model posits that each participant has a unique, inherent discount rate. Within this framework, the following softmax function was utilized to transform the difference subjective values of the two options ($V_{LL} - V_{SS}$) on each trial into the probability of selecting the delayed option:

$$P_{LL} = \frac{1}{1 + e^{-T(V_{LL} - V_{SS})}} \tag{2}$$

where $T$ represents the inverse temperature parameter specific to each participant, characterizing the variability or randomness in a person's decision-making process. A lower value of $T$ leads to increased nonsystematic fluctuations around the point of indifference, which is the point where both options are equally favored. During the *Self1* block of the experiment, the free parameter $k$ was assigned values ranging from −4 to 0, while the $\log_{10}(T)$ parameter (denoted as $t$) had its value set within a range from −1 to 1.

**Preference-uncertainty (KU) model.** Contrary to the KT model described earlier, the KU model suggests that participants' discount rates should be viewed as a distribution, rather than a single definitive value [34]. On each trial, participants draw a $k$ value from a normally

distributed discounting distribution that is specific to each participant and is updated after every trial:

$$P_k = \mathcal{N}\left(k; km, ku^2\right)$$

(3)

where free parameters $km$ and $ku$ correspond to the mean and standard deviation of the normal distribution, respectively. Derived from Eq (1), participants will only choose the delayed option under the condition that $k < \log_{10}\left[(M_{LL}/M_{SS} - 1)/D\right]$; the probability of selecting the delayed option, given a single sampled value from the discounting distribution specified in Eq (3) is:

$$P_{LL} = \Psi\left(\log_{10}\left[(M_{LL}/M_{SS} - 1)/D\right]; km, ku^2\right)$$

(4)

where $\Psi$ represents the cumulative distribution function of the normal distribution. Model fitting was conducted using R [119] (v4.2.1), Stan [125] (v2.32), and the RStan package [126] (v2.21.7). We employed Hamilton Monte Carlo (HMC), an advanced and efficient Markov Chain Monte Carlo (MCMC) sampling method.

Our study focused on testing the involvement of mPFC in people's susceptibility to social influence. Building upon our previous work [36], we employed the established KU model as our analytical framework to assess data from these lesion participants. We successfully recovered all the parameters in the KU model (all $r_s > 0.87$, S1 Fig) as well as confirming excellent posterior predictive accuracy of the modeled parameters (S2 Fig).

## Model fitting

We used R (v4.2.1), Stan (v2.32), and the RStan package (v2.21.7) for model fitting. Stan makes use of HMC, an exceptionally efficient MCMC sampling method, to perform full Bayesian inference and accurately determine the true posterior distribution. We applied hierarchical Bayesian modeling to analyze participants' decisions on a trial-by-trial basis. In hierarchical Bayesian modeling, the individual-level parameter, denoted by $\phi$, was sampled from a group-level normal distribution, as follows:

$$\phi \sim \mathcal{N}\left(\mu_\phi, \sigma_\phi^2\right)$$

(5)

where $\mu_\phi$ and $\sigma_\phi$ represent the group-level mean and standard deviation, respectively. The group-level parameters were defined using weakly-informative priors: $\mu_\phi$ followed a normal distribution centered around 0, with a standard deviation that was adjusted based on free parameters. Concurrently, $\sigma_\phi$ was modeled using a half-Cauchy distribution, with its location parameter set at 0 and its scale parameter adjusted in accordance with free parameters. In the KT model, the parameter $k$ was subjected to a negative constraint, whereas $t$ was constrained to lie within the range of $[-1, 1]$. In the KU model, the parameter $km$ was negatively constrained, whereas $ku$ was constrained positively. To facilitate more conservative estimation of all free parameters, priors were reset at the start of each experimental block. Hierarchical Bayesian modeling was applied separately for the groups of HCs, mPFC lesion patients, and LCs, with identical weakly-informative priors used across groups to promote conservative parameter estimation [127,128].

All free parameters at both the group and individual levels were simultaneously estimated through Bayes' theorem by integrating behavioral data. We fitted each model with four independent HMC chains, where each chain included 2,000 iterations following an initial 2,000

warm-up iterations. This process generated a total of 8,000 valid posterior samples. The convergence of HMC was assessed both visually by examining trace plots, and quantitatively by using the Gelman-Rubin $\hat{R}$ statistics. In the chosen model, the $\hat{R}$ values for all free parameters were close to 1.0, indicating that convergence was achieved satisfactorily.

## Parameter recovery

Following model fitting, we verified the identifiability of parameters through parameter recovery. Let $\phi$ denotes a generic free parameter in the selected model. We randomly drew a set of group-level parameters from the identical weakly-informative prior group-level distribution that was used in model fitting. Here, $\mu_\phi$ and $\sigma_\phi$ represent the mean and standard deviation at the group level, respectively:

$$\mu_\phi \sim \mathcal{N}(0,\ 3)$$

$$\sigma_\phi \sim \mathcal{HC}(0,\ 2) \tag{6}$$

where $\mathcal{HC}$ refers to the half-Cauchy distribution. Next, we generated data for 120 synthetic participants by deriving their parameters from this set of group-level parameters. For these 120 synthetic participants, their individual-level parameters, denoted as $\phi_i$, were drawn from a normal distribution using the corresponding group-level parameters:

$$\phi_i \sim \mathcal{N}\left(\mu_\phi, \sigma_\phi^2\right) \tag{7}$$

Subsequently, we employed the chosen model as a tool to generate simulated behavioral data for our social discounting task. Specifically, we simulated decisions across 50 trials for each synthetic participant, using the choice pairs derived from the generative method (see the below *Optimization of choice pairs*). Then, we applied our selected model to the simulated data following the same procedure we used for the actual participant data. Particularly, we fitted the KU model to the individual simulated data using HMC through Stan. This process resulted in posterior distributions for the free parameters at both group and individual levels. Finally, we calculated Spearman's Rho correlations to compare the simulated and recovered parameters at the individual level. We repeated the entire parameter recovery process 20 times, averaging the Spearman's Rho correlation coefficients through Fisher's Z-transformation.

## Posterior predictive checks

We used posterior predictive checks to assess how well the posterior estimates from our winning model replicated key aspects of participants' behavior, such as their ability to learn others' preferences. Specifically, we employed a posthoc absolute-fit approach [54], which took into account participants' actual decisions and option pairs, to generate predictions using the entire set of posterior MCMC samples from the winning model. We generated synthetic decisions repeatedly, matching the number of MCMC samples (i.e., 8,000 times) for each trial and each participant, using individual-level posterior parameters obtained from model estimation. We then analyzed the synthetic data with the same methods applied to the actual data, using a LMM. This LMM included fixed effects of group (*HCs*, *mPFC lesions*, and *LCs*), other's preference (*patient* versus *impulsive*), and their interactions, along with a random subject-level intercept.

## Optimization of choice pairs

To accurately estimate participants' preferences for discounting, choice pairs in all *Self* blocks were generated by switching between two methods: generative and adaptive methods, within the context of the KT model framework. The generative approach entailed creating every possible pair of amounts and delays for the choice options. Within each *Self* block, 25 trials (i.e., half of the trials in each *Self* block) were selected to closely match the indifference points of 25 hypothetical participants. These participants had $k$ values that were uniformly distributed across the range from −4 to 0. This method provided an efficient yet somewhat imprecise estimation of participants' discounting parameters. The other 25 trials in each *Self* block were created through an adaptive approach, utilizing a Bayesian framework to achieve precise estimates of the discounting parameters [129,130]. Previous studies have shown that this technique can generate more reliable estimates of the $k$ value with fewer trials needed. The participant's initial prior belief about $k$ was defined as a normal distribution with a mean of −2 and a standard deviation of 1, and $t$ was fixed at 0.3. After every decision by the participant, their belief distribution of $k$ was updated according to Bayes' theorem. Following this update, choice pairs were generated to test our estimate of the participant's indifference point, derived from the expected value of $k$'s current posterior distribution.

For all *Other* blocks and parameter recovery processes, choice pairs were exclusively generated using the generative method. The choices given to participants were specifically structured to match the indifference points of 50 hypothetical participants, whose $k$ values were evenly spread from −4–0.

## Simulation of the other people's choices

The behaviors of the two other people were modeled based on the participants' baseline discount rates, which were determined through the KT model during the *Self1* block. More specifically, the decisions of the other people were generated by a simulated hyperbolic discounting model, where the discount rate $k$ was adjusted to be either plus one (*more impulsive*) or minus one (*more patient*) from the participant's own baseline $k$ in the first experimental block. Importantly, the decisions made by the simulated hyperbolic discounter were subject to an extent of randomness. This randomness arose from the process of converting the subjective value of options into a choice probability through a softmax function with the inverse temperature parameter $t = 1$. The order of the other people's preferences (*more impulsive* versus *more patient*) was counterbalanced across participants.

## Signed Kullback–Leibler divergence

The $D_{KL}$, which quantifies the difference between two probability distributions [55], was used to measure the variation in participants' discount rates ($k$) after learning about the other people. $D_{KL}$ is defined as follows:

$$D_{KL}(P\|Q) = \int_{-\infty}^{\infty} p(x)\log_{10}\left(\frac{p(x)}{q(x)}\right) dx$$

(8)

where $P$ and $Q$ represent the distributions of a continuous random variable over a sample space, $\mathcal{X}$, and $p$ and $q$ denote the respective probability densities of $P$ and $Q$. In our study, we used $D_{KL}$ to quantify the divergence between the posterior distributions of $k$ at the end of two successive *Self* blocks. $D_{KL}$ was signed for subsequent analyses [35]. Positive signed $D_{KL}$ values indicate a shift in participants' discounting preferences toward those of the other people,

whereas negative signed $D_{KL}$ values suggest a move away from the other people's preferences, relative to the baseline discounting preferences:

$$\text{Signed } D_{KL} = \begin{cases} D_{KL}, & if \ \frac{km_{\text{other}, \ i} - km_{\text{self}, \ 1}}{km_{\text{self}, \ i+1} - km_{\text{self}, \ 1}} > 0 \\ -D_{KL}, & if \ \frac{km_{\text{other}, \ i} - km_{\text{self}, \ 1}}{km_{\text{self}, \ i+1} - km_{\text{self}, \ 1}} < 0 \end{cases} \tag{9}$$

where $km$ represents the mean of the discount rate distribution as estimated by the KU model, and the subscript $i$ indicates the number of *Other* blocks (i.e., either 2 or 4). For instance, if a participant's discounting preference becomes more negative (i.e., more patient) following exposure to the discounting preference of a more patient other person, this change would be reflected by a positive signed $D_{KL}$ value. On the other hand, negative signed $D_{KL}$ values indicate that the participant's discounting preferences have diverged from those of the other people.

## Voxel-based lesion-symptom mapping (VLSM)

Two behavioral regressors of interest were selected for VLSM based on our a priori hypotheses:

1. Contrasts between susceptibilities to impulsive and patient social influence (i.e., signed impulsive $D_{KL}$ – signed patient $D_{KL}$)

2. Self baseline discount rates (i.e., self $km$ in the *Self1* block)

The examination of the contrasts between susceptibilities to impulsive and patient social influence aimed to determine if damage to specific subregions of the mPFC was responsible for the increased susceptibility to impulsive social influence observed in the between-group analysis. This analysis only included participants who had both patient and impulsive others present. Additionally, we tested whether the heightened temporal impulsivity observed in the mPFC lesion group, compared to HCs, was linked to distinct subregions of the mPFC.

We utilized FSL [58] (v6.0.7.6)'s *randomize* function to conduct a permutation-based VLSM analysis [59,60], which compares lesion participants with damage at each voxel to all other lesion participants. FSL has been validated for performing VLSM analyses and is widely utilized, as highlighted by its adoption in several recent lesion studies [131–135]. FSL implements the latest advancements in brain-based analysis, maintaining regular updates, and remaining open source. FSL also supports the use of TFCE, which maximizes power and uses nonarbitrary definitions of cluster size [59]. This feature is not currently available in other lesion-mapping toolboxes, such as LESYMAP and NiiStat. To increase power, we mirrored the lesion participants' lesion maps, as we did not have specific hypotheses about laterality of mPFC function [134,135], resulting in symmetrical masks. Voxels were included in the VLSM analysis only if at least five participants had some degree of damage in that voxel. Each behavioral regressor of interest was ranked to correct for skewness in the residuals distribution [60] and then $z$-scored, in accordance with the requirements of FSL to align with the nature of our experimental design, before being input into the FSL design files.

*P* values were generated through permutation-based TFCE in randomize with 5,000 permutations and FSL's default TFCE settings, which are optimized for this type of data [59,60]. Permutation testing repeats the same analysis multiple times with the randomly shuffled data to calculate voxel-wise *P* values, which estimate the probability that the observed effect could be attributed to random noise. This approach therefore more accurately reflects the nature of the data, relies on fewer assumptions compared to other methods, and can be combined with

the advantages of TFCE [60]. Permutation testing is widely recognized as the 'gold standard' for addressing multiple comparisons in VLSM studies [136]. By combining permutation testing with TFCE, we effectively balanced sensitivity to true effects while minimizing the likelihood of detecting small, potentially spurious effects [56]. To ensure even greater stringency, we further applied a Bonferroni correction for multiple comparisons across the two behavioral regressors of interest ($p < 0.025$) to the uncorrected maps from the permutation-based TFCE results. For the purpose of visualization, we applied binarized masks of the significant areas from each analysis to the $t$-values.

## Supporting information

**S1 Text. Temporal impulsivity and preference uncertainty do not depend on depression or apathy levels.**
(PDF)

**S1 Table. Summary of demographic variables for each group and linear regression.**
(PDF)

**S2 Table. Linear mixed-effects model predicting learning performances.**
(PDF)

**S3 Table. Linear mixed-effects model predicting susceptibility to social influence, with self baseline temporal impulsivity as covariates (centered around the grand mean).**
(PDF)

**S4 Table. LMM predicting susceptibility to social influence, with self baseline temporal impulsivity as covariates (centered around the grand mean), controlling for the order of others' preferences.**
(PDF)

**S5 Table. Correlations between impulsive and patient signed KL divergence (DKL).**
(PDF)

**S6 Table. LMM predicting susceptibility to social influence, with self baseline temporal impulsivity km as covariates (centered around the grand mean), controlling for self baseline preference uncertainty ku (centered around the grand mean).**
(PDF)

**S7 Table. Correlations between learning performances and signed KL divergence (DKL).**
(PDF)

**S8 Table. Correlations between perceived similarity and signed KL divergence (DKL).**
(PDF)

**S1 Fig. Parameter recovery.** The confusion matrix illustrates Spearman's Rho correlations between simulated and recovered (fitted) parameters. Both km and ku showed robust positive correlations between their true and recovered values, with all rs >0.87.
(TIF)

**S2 Fig. Posterior predictive checks of the winning model.** Posterior prediction replicates the key patterns observed in our empirical data. All three participant groups (healthy controls, mPFC lesions, and lesion controls) successfully learned the task (right-tailed exact binomial tests against 50%, all ps < 0.001). Compared to healthy controls, both mPFC lesion patients and lesion controls showed less accuracy in learning others' preferences, regardless of

whether these preferences were impulsive or patient (main effect mPFC vs. HC, b [95% CI] = −4.04 [−5.99 −2.09], $p < 0.001$; main effect LC vs. HC, b [95% CI] = −4.32 [−6.82 −1.83], $p < 0.001$). Participants generally performed better in terms of learning the preferences of patient others than impulsive ones (main effect patient vs. impulsive, b [95% CI] = 1.89 [0.95, 2.83], $p < 0.001$). Large bordered circles indicate the mean, error bars show the standard error of the mean, dots represent raw simulated data, and asterisks denote significant main effects of groups from the linear mixed-effects model. Note that the vertical axis starts at 50%, representing the chance level. **$p < 0.001$. Red dots are the means of actual data. (TIF)

**S3 Fig. Damage to dmPFC (area 9) enhances susceptibility to impulsive social influence, including both mPFC lesion participants and lesion controls.** **(a)** Permutation-based, whole-brain, nonparametric voxel-based lesion-symptom mapping (VLSM) showed that damage to dorsomedial prefrontal cortex (dmPFC, area 9, peak MNI coordinate [±2, 40, 20]) was correlated with enhanced susceptibility to impulsive relative to patient social influence (permutation-based threshold free cluster enhancement (TFCE) $p < 0.025$). **(b)** Plotting the ranked contrasts between susceptibilities to impulsive and patient social influence, separately for participants with lesions or no lesion in this area identified by the VLSM analysis. $N = 42$ for this analysis where both patient and impulsive others were present. The underlying data and code used to generate this figure can be found at https://osf.io/qzurp/. Note: panel (B) is for illustrative purposes only and displays the ranked difference in signed KL divergence contrasts between participants with vs. without lesions, in the ROI defined by a wholebrain contrast. (TIF)

**S4 Fig. Damage to vmPFC and ventral striatum increases temporal impulsivity, including both mPFC lesion participants and lesion controls.** **(a)** Permutation-based, whole-brain, nonparametric voxel-based lesion-symptom mapping (VLSM) showed that the area 25 in the vmPFC as well as ventral striatum where damage was correlated with heightened temporal impulsivity (permutation-based threshold free cluster enhancement (TFCE) $p < 0.05$). **(b)** Plotting the ranked self baseline discounting preferences, 165 separately for participants with damage or no damage in the areas identified by the VLSM analysis (N 1= 50 in total). The underlying data and code used to generate this figure can be found at https://osf.io/qzurp/. Note: panel (B) is for illustrative purposes only and displays the ranked difference in self baseline discounting preferences between participants with vs. without lesions, in the ROI defined by a wholebrain contrast. (TIF)

## Acknowledgments

We would like to thank Ayat Abdurahman, Daniel Drew and Luca Hargitai for assistance with data collection. We would also like to thank Andrea Reiter and Michael Moutoussis for their useful advice regarding the Bayesian computational modeling and Matthew Apps, Louisa Thomas and Joshua Balsters for useful discussions.

## Author contributions

**Conceptualization:** Mona M. Garvert, Sanjay G. Manohar, Patricia L. Lockwood.

**Formal analysis:** Zhilin Su, Mona M. Garvert, Lei Zhang, Todd A. Vogel, Jo Cutler.

**Funding acquisition:** Masud Husain, Patricia L. Lockwood.

**Investigation:** Patricia L. Lockwood.

**Methodology:** Mona M. Garvert, Sanjay G. Manohar, Patricia L. Lockwood.

**Supervision:** Todd A. Vogel, Patricia L. Lockwood.

**Visualization:** Zhilin Su.

**Writing – original draft:** Zhilin Su, Patricia L. Lockwood.

**Writing – review & editing:** Zhilin Su, Mona M. Garvert, Lei Zhang, Todd A. Vogel, Jo Cutler, Masud Husain, Sanjay G. Manohar, Patricia L. Lockwood.

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
