## [Editor Report · Decision Letter 0]

12 Nov 2024

Dear Dr Su,

Thank you for submitting your manuscript entitled "Dorsomedial and ventromedial prefrontal cortex lesions differentially impact social influence and temporal impulsivity" for consideration as a Research Article by PLOS Biology.

Your manuscript has now been evaluated by the PLOS Biology editorial staff as well as by an academic editor with relevant expertise and I am writing to let you know that we would like to send your submission out for external peer review.

Once your full submission is complete, your paper will undergo a series of checks in preparation for peer review. After your manuscript has passed the checks it will be sent out for review. To provide the metadata for your submission, please Login to Editorial Manager (https://www.editorialmanager.com/pbiology) within two working days, i.e. by Nov 14 2024 11:59PM.

Kind regards,

Christian

Christian Schnell, PhD

Senior Editor

PLOS Biology

cschnell@plos.org

---

## [Decision Letter · Decision Letter 1]

10 Jan 2025

Dear Dr Su,

Happy New Year!

Thank you for your patience while your manuscript "Dorsomedial and ventromedial prefrontal cortex lesions differentially impact social influence and temporal impulsivity" went through peer-review at PLOS Biology. Your manuscript has now been evaluated by the PLOS Biology editors, an Academic Editor with relevant expertise, and by several independent reviewers.

In light of the reviews, which you will find at the end of this email, we are pleased to offer you the opportunity to address the [comments/remaining points] from the reviewers in a revision that we anticipate should not take you very long. We will then assess your revised manuscript and your response to the reviewers' comments with our Academic Editor aiming to avoid further rounds of peer-review, although might need to consult with the reviewers, depending on the nature of the revisions.

**IMPORTANT - SUBMITTING YOUR REVISION**

*Resubmission Checklist*

*Published Peer Review*

*PLOS Data Policy*

*Blot and Gel Data Policy*

Sincerely,

Christian

Christian Schnell, PhD

Senior Editor

PLOS Biology

cschnell@plos.org

REVIEWS:

Reviewer #1: Su et al. present a lesion-based investigation of the role of medial prefrontal cortex in temporal impulsivity and (receiving) social influence. Participants, including those with MPFC lesion, other lesions, and healthy controls, completed a temporal discounting task for themselves, and simulated more impulsive and more patient others. All groups of participants learned the idiosyncratic temporal preferences of both simulated others. Healthy controls were less impulsive than the lesion groups at baseline, and the non-MPFC lesion control group expressed less certain preferences at baseline. The MPFC lesion group demonstrated higher overall influence of the simulated others' preferences on their own subsequent intertemporal choices, with a particularly strong effect for the impulsive (vs. patient) other. VLSM identified a region in DMPFC associated with this effect. In contrast, lesions in VMPFC and the ventral striatum were associated with higher baseline impulsivity. Together, these results implicate MPFC - and in particular, DMPFC - in receptivity to social influence.

Generally, I think this is a strong paper, featuring a well-designed paradigm, sensible analyses, and an interesting research question, and clear writing. Below I list a few questions and comments I hope might improve it further:

1. Perhaps I am misunderstanding, but in figures 5 and 6, I am concerned that the "b" panels are non-independent. That is, they show the ranked difference in KL divergence/baseline impulsivity between participants with vs. without lesions in the ROI defined by a wholebrain contrast. I realize that these panels do not reflect an inferential analysis, but this kind of circularity still exaggerates the size of the effect.

2. Why don't the linear mixed effects models include any random slopes? As currently written, the models assume that the size of the patient vs. impulsive effect on KL divergence is the same across participants in the same group, which isn't necessarily true.

3. Eyeballing figure 4, it looks like there might be a negative correlation across participants within the same group, such that those who are more influenced by the impulsive other are less influenced by the patient other, and vice versa. Is that true? And if it is, is that a substantive effect, or is it just something like an order effect?

4. While I agree that lesion evidence offers strong evidence for causation than neuroimaging, the lesions are obviously not randomly assigned, so the causal identification that they license isn't absolute. For example, if hypertension and impulsivity share a common cause, which doesn't seem impossible, then this could account for the association between lesions and baseline impulsivity through an alternative causal pathway.

5. It is interesting that the impulsive others had a bigger influence on the lesion participants, given that all groups of participants learned more accurately about the patient others. Was there any association between participants' learning accuracy and the degree to which they were influenced by the others?

6. I could not directly review the data/code. The statement before the beginning of the manuscript states that they are fully available, but instead of an actual link, there is just a placeholder text reading "[URL]".

Reviewer #2: In this paper, the authors leveraged access to a relatively large sample of patients with damage to the mPFC to examine how the structural integrity of different regions within mPFC relate to temporal discounting, as well as to social influence. The authors replicated the finding that mPFC damage leads to increased discounting, but they built on this finding by pinpointing the exact (ventral) subregions associated with increased discounting. The authors also found novel evidence that the ventral striatum may play a causal role in future-oriented choice, although they do emphasize the exploratory nature of that finding. Finally, the authors provide compelling evidence that the dorsal portion of the mPFC is causally involved in maintaining one's own preferences as separate from another person's. While I'm less familiar with the literature on social influence, I believe that this is a pretty novel finding as well.

I really enjoyed this paper, so I don't have many comments. The analyses are sound, the results are interesting, and the writing is very clear. I just have a few small suggestions:

- The authors do a nice job in the Discussion speculating about why dmPFC damage might lead to increased susceptibility to social influence. I think the hypothesis that the dmPFC maintains a self-other distinction in some way makes sense. However, there are a couple of findings related to this that I would have wanted to see discussed as well. First, why do the authors think that the social influence results are driven by the impulsive condition rather than the patient condition? Could the authors comment on this asymmetry in the direction of social influence? Second, the authors bring up that their results might have to do with social learning, but they also found that there was no evidence that learning was associated with the extent of social influence. It's worth mentioning that null finding there as a caveat.

- In line 267, the authors mention that participants with mPFC lesions "did not report feeling more similar to impulsive others." When I first read this (having not read the Methods section beforehand), I was confused about what this meant. Could the authors add a sentence to the first part of the Results section (and/or to the caption of Figure 2) just mentioning that the participants were also asked about their perceived similarity to the other choosers?

- I know that examining preference uncertainty was not the main goal of the study, but I'm intrigued by the finding that the participants with lesions outside the mPFC had more preference uncertainty. Would the authors consider doing an exploratory VLSM analysis to see where in the brain more damage is associated with more preference uncertainty?

- The authors (appropriately) control for baseline temporal discounting in their models that examine social influence. However, one could argue that controlling for baseline preference uncertainty is also important, in light of the fact that the LC group showed higher preference uncertainty than the other groups. Preference uncertainty also seems like it might be related to susceptibility to social influence; I could imagine that people whose preferences are more flexible or uncertain might be more susceptible. Could the authors run an additional analysis examining social influence that includes preference uncertainty as a covariate?

- Finally, I suggest that the authors consider revising the title so that it says "temporal discounting" instead of "temporal impulsivity." Impulsivity can refer to a number of different constructs, while "temporal discounting" more specifically refers to the phenomenon of overvaluing more immediate rewards at one's future expense.

Reviewer #3: In this manuscript, Su and colleagues investigate the impact of prefrontal cortex lesions on individuals' temporal impulsivity and their sensitivity to social influence. The experimental design consists in a set of inter-temporal decision tasks, proposed to a relatively large cohort of lesioned patients, lesion controls and healthy controls. Participants alternate classical inter-temporal decision blocks, and blocks where they have to learn the temporal preferences of two other (simulated) people (one more impulsive and one more patient). The authors use Bayesian computational models to estimate baseline discounting and susceptibility to social influence, measured as the change in discounting after learning others' economic preferences. They then use structural neuroimaging (voxel-based lesion-symptom mapping) to link behavioral effects of lesions to specific area in the brain, and claim to demonstrate that the dmPFC is causally implicated in susceptibility to social influence, with distinct ventral portions of mPFC involved in temporal discounting.

This manuscript represents a laudable effort to investigate the question of interest, with data collection on a relatively large cohort of rare patients, state of the art Bayesian models and statistical analyses, etc. Nonetheless, I feel that severe experimental and analytical limitations weaken the solidity of the claims made in the current version of the manuscript. I hope the authors will find my comments useful to amend their claims.

Main issues

1) Although I acknowledge that similar paradigms have been used in the past, I am not convinced at all that the current experimental paradigm allows to make solid claim on "social influence". Basically, the only "social" aspect of the task is the fake surnames displayed on the Others block. There is no non-social condition, such that it is entirely possible that the effects are purely driven by non-social instrumental associative learning.

2) Although the task is elegant and sophisticated in the way it tries to create individual specific condition, based on individual estimated discounting rate, this also create a fundamental problem of comparability. Basically, given that the different group (lesion, lesion control and healthy controls) have different discount rates, they then face different social-influence/learning tasks, with different Others, such that isolating the effect of Groups on social influence seems virtually impossible. Maybe developping a model that explicit model the learning could start addressing this issue.

3) L 255-257: one of the main behavioral result is a close to threshold (p = 0.031) interaction, arising from a complex mixed-model modelling factorial effects on computational parameters. How robust is this effect to different specifications of the LLM (e.g. including random slopes, or different set of covariates)?

4) A significant portion of the neuroimaging results are based on what seems to be extremely lenient statistical threshold (P< 0.05 uncorrected). While I acknowledge that the few reported voxels land in regions "of interest", I do not feel that these results build on solid statistical evidence, and I even more strongly urge against the idea of reporting them as one of the main claim/finding in the abstract.

---

## [Decision Letter · Decision Letter 2]

14 Feb 2025

Dear Dr Su,

Thank you for your patience while we considered your revised manuscript "Dorsomedial and ventromedial prefrontal cortex lesions differentially impact social influence and temporal discounting" for publication as a Research Article at PLOS Biology. This revised version of your manuscript has been evaluated by the PLOS Biology editors, the Academic Editor and two of the original reviewers.

Based on the reviews and on our Academic Editor's assessment of your revision, we are likely to accept this manuscript for publication, provided you satisfactorily address the remaining reviewer comments via textual revisions in the discussion and the following data and other policy-related requests:

* All research involving human participants must have been approved by the authors' Institutional Review Board (IRB) or an equivalent committee, and must have been conducted according to the principles expressed in the Declaration of Helsinki. Please provide this information in the Methods section and also provide the approval numbers for this study.

* Please move the supplementary methods and the reference to the main manuscript. We have no restrictions on word count or the number of references.

* DATA POLICY:

Regardless of the method selected, please ensure that you provide the individual numerical values that underlie the summary data displayed in the following figure panels as they are essential for readers to assess your analysis and to reproduce it: 2C, 3B, 4, 5B, 6B, S2, S3B and S4B.

* Please ensure that you are using best practice for statistical reporting and data presentation. These are our guidelines https://journals.plos.org/plosbiology/s/best-practices-in-research-reporting#loc-statistical-reporting and a useful resource on data presentation https://journals.plos.org/plosbiology/article?id=10.1371/journal.pbio.1002128

* If you are reporting experiments where n ≤ 5, please plot each individual data point.

* CODE POLICY

We expect to receive your revised manuscript within two weeks.

*Published Peer Review History*

*Press*

Sincerely,

Christian

Christian Schnell, PhD

Senior Editor

cschnell@plos.org

PLOS Biology

Reviewer remarks:

Reviewer #1 (Mark A. Thornton): The authors have very thoroughly addressed the issues I raised in my initial review, which were for the most part minor to begin with. I think this paper will make a substantial contribution to the literature in its present form.

Reviewer #3: I thank the authors for their serious revision and constructive attitude. While I consider my concerns #3 and #4 satisfactorily addressed, I am not convinced by the arguments provided to rebut my concerns #1 and #2. Although it is not my intention to engage into a long and sterile back and forth revision process - let's agree to disagree -, let me restate -for the records- what I think are clear limitations that seriously question the value, robustness and generalizability of the claims proposed by the authors:

#1: I do not question the fact that participant do learn to adapt to the feedback that is proposed (about the fictitious participants' preference), but raise the (very credible IMHO) possibility that this learning (or a significant portion of it) can be done without mobilizing any social cognitive process. The same behavior could probably be observed if the task was to adjust to an explicit algorithm or bandit. The fact that the lab routinely engage in similar deceptive procedure seriously weaken the believability counter-argument. The incentivization argument is also not very strong, as participants are known to be very driven by intrinsic reward (solving a task, be correct) versus extrinsic rewards (meagre monetary bonuses). Hence, I sincerely doubt the very general claims about social influence.

#2: I still think that the difference in the task performed by the subject is a serious problem in how we interpret the findings. Imagine a similar procedure in the realm of simple reinforcement-learning (RL) task. First, we want to control for the difference in how two groups value the outcome, and realize that Group 1 need 10£ to reach the same behavior as Group 2 with 1£ in a decision-making task. Then, the two groups perform a RL task (group 1 with 10£ rewards and Group 2 with 1£ rewards), and the authors report a significant difference in learning-rate between the groups: how can we draw a clear inference and interpretation of this finding ?

---

## [Editor Report · Decision Letter 3]

21 Feb 2025

Dear Zhilin,

Thank you for the submission of your revised Research Article "Dorsomedial and ventromedial prefrontal cortex lesions differentially impact social influence and temporal discounting" for publication in PLOS Biology. On behalf of my colleagues and the Academic Editor, Raphael Kaplan, I am pleased to say that we can in principle accept your manuscript for publication, provided you address any remaining formatting and reporting issues. These will be detailed in an email you should receive within 2-3 business days from our colleagues in the journal operations team; no action is required from you until then. Please note that we will not be able to formally accept your manuscript and schedule it for publication until you have completed any requested changes.

PRESS

Sincerely, 

Christian

Christian Schnell, PhD

Senior Editor

PLOS Biology

cschnell@plos.org